# Comparative Analysis of Ease of Removal of Fractured NiTi Endodontic Rotary Files from the Root Canal System—An In Vitro Study

**DOI:** 10.3390/ijerph19020718

**Published:** 2022-01-10

**Authors:** Vicente Faus-Matoses, Eva Burgos Ibáñez, Vicente Faus-Llácer, Celia Ruiz-Sánchez, Álvaro Zubizarreta-Macho, Ignacio Faus-Matoses

**Affiliations:** 1Department of Stomatology, Faculty of Medicine and Dentistry, University of Valencia, 46010 Valencia, Spain; vicente.faus@uv.es (V.F.-M.); eburi@alumni.uv.es (E.B.I.); fausvj@uv.es (V.F.-L.); ceruizsan@gmail.com (C.R.-S.); ignacio.faus@uv.es (I.F.-M.); 2Department of Endodontics, Faculty of Health Sciences, Alfonso X El Sabio University, 28691 Madrid, Spain; 3Department of Surgery, Faculty of Medicine and Dentistry, University of Salamanca, 37008 Salamanca, Spain

**Keywords:** endodontics, cyclic fatigue, endodontic rotary file removal, dentin removal, ultrasonics, Endo Rescue

## Abstract

This study aimed at analyzing and comparing the ease of removal of fractured nickel–titanium (NiTi) endodontic rotary files from the root canal system between the ultrasonic tips and the Endo Rescue appliance removal systems, as well as comparing the volume of dentin removed between ultrasonic tips and the Endo Rescue appliance using a micro-computed tomography (micro-CT) scan. Material and Methods: Forty NiTi endodontic rotary files were intentionally fractured in 40 root canal systems of 20 lower first molar teeth and distributed into the following study groups: A: Ultrasonic tips (*n* = 20) (US) and B: Endo Rescue device (*n* = 20) (ER). Preoperative and postoperative micro-CT scans were uploaded into image processing software to analyze the volumetric variations of dentin using an algorithm that enables progressive differentiation between neighboring pixels after defining and segmenting the fractured NiTi endodontic rotary files and the root canal systems in both micro-CT scans. A non-parametric Mann–Whitney–Wilcoxon test or *t*-test for independent samples was used to analyze the results. Results: The US and ES study groups saw 8 (1 mesiobuccal and 7 distal root canal system) and 3 (distal root canal system) fractured NiTi endodontic rotary files removed, respectively. No statistically significant differences were found in the amount of dentin removed between the US and ER study groups at the mesiobuccal (*p* = 0.9109) and distal root canal system (*p* = 0.8669). Conclusions: Ultrasonic tips enable greater ease of removal of NiTi endodontic rotary files from the root canal system, with similar amounts of dentin removal between the two methods.

## 1. Introduction

Nickel–titanium (NiTi) alloy endodontic rotary files are widely used to clean and shape the root canal system because they have been shown to significantly increase the success rate of root canal treatments [1]. However, the unexpected fracture of NiTi alloy endodontic rotary files during the root canal treatment can sometimes influence outcomes [2], with the incidence of fracture of NiTi alloy endodontic rotary files estimated between 0.4% to 4.6% [3,4] by torsional fatigue or bending fatigue [5]. Several procedures can be used to remove fractured instruments [6], including the Masseran kit [6], Endo Safety System [7], and Endo Extractor [8]. Additionally, other techniques have been proposed for the removal of fractured NiTi alloy endodontic rotary files, including the wire loop technique [9], spinal tap needle-and-Hedstrom file technique [10], Cancelliers [11], Tube-and-Hedstrom file technique [12], hypodermic needle [13], blunt needle and core paste technique [14], and the Instrument Removal System (IRS) [15]. However, all these procedures require that the coronal third of the root canal system be adequately enlarged to provide access to the fractured instruments [1]. This implies a loss of dentin tissue, which can negatively affect the structural integrity of the tooth [16]. Furthermore, it can result in perforated roots or potentially put teeth at risk of vertical root fracture, particularly in the apical third [1]. Additionally, the use of magnification in combination with ultrasonic has been widely recognized as the best technique for removing separated instruments, especially the technique proposed by Ruddle, who recommended preparing a platform in the pre-enlarged root canal system using ultrasonic tips. The removal of the fractured instruments can be performed using a range of techniques, many of them employing some type of microtube since the direct application of ultrasonic energy cannot remove fractured instruments [1]. As a result, there is no universal procedure for removing fractured instruments, and removals are often performed using an array of techniques and devices, with inherent risks and limitations.

Today, micro-computed tomography (micro-CT) scans are used to study root canal anatomy and assess changes in root canal morphology after root canal treatment [17,18,19]. Madarati et al. used a micro-CT scan to analyze changes in the root canal space after removal of fractured instruments in canine teeth using ultrasonics [20]. However, many studies have shown that the separation of fractured endodontic instruments occurs mostly in curved and narrow root canal systems, such as the mesiobuccal root canal of upper molars or the mesial root canal of lower molars, due to their complex anatomy and curved canal structure [10].

The objective of the present study was to assess and compare the ease of removal of fractured NiTi alloy endodontic rotary files from the root canal system between the ultrasonic tips and Endo Rescue appliance removal systems, as well as to compare the volume of dentin removed between ultrasonic tips and Endo Rescue appliance using a micro-CT scan. The null hypothesis is (H_0_) that there is no difference in ease of removal of fractured NiTi endodontic rotary files between the ultrasonic tips and Endo Rescue appliance, and there is no difference in the dentin removal of fractured NiTi alloy endodontic rotary files between ultrasonic tips and Endo Rescue appliance.

## 2. Materials and Methods

### 2.1. Study Design

Forty root canal systems (20 mesial and 20 distal) extracted from twenty lower first molar teeth for periodontal reasons were selected for this study. The systems had mature roots and no incidence of previous root canal treatment, root resorption, or calcium metamorphosis. The study was carried out from February to October 2021 at the Department of Stomatology of the University of Valencia (Valencia, Spain). The study was a randomized controlled experimental trial in accordance with the statement by the German Ethics Committee (Zentrale Ethikkommission, 2003) on the use of organic tissues in medical research. The study design was approved by the Ethics Committee of Alfonso X El Sabio University (Process no. 24/2020). All patients provided their informed consent to transfer their teeth for the study.

### 2.2. Experimental Procedure

Digital preoperative radiographs were taken buccolingually and mesiodistally to analyze the root canal system anatomy of the teeth selected for the study. A single operator then performed the endodontic access cavities using the technique outlined by Rover et al. [21]. The working length of the root canal was established with a stainless steel #10 K-file (Dentsply Maillefer, Ballaigues, Switzerland) inserted until the tip became visible through the apical foramen. The canals were prepared using a Protaper Gold endodontic rotary system (Dentsply Maillefer, Ballaigues, Switzerland) and endodontic rotary file up to F1. They were irrigated with 5 mL of 5.25% NaOCl (Clorox; Oakland, CA, USA), 5 mL of sterile saline solution (Braun, Jaén, Spain), and 5 mL of 17% EDTA (SmearClear; SybronEndo, CA, USA) using an endodontic needle (Miraject Endo Luer; Hager & Werken, Duisburg, Germany) with an apical diameter of 0.3 mm inserted up to 1 mm of the working length. The apical 4 mm of the F2 endodontic rotary files (Protaper Gold, Dentsply Maillefer, Ballaigues, Switzerland) was then partially cut with a low-speed 0.3 diamond disk (Brasseler, GA, USA, Savannah, GA, USA) and intentionally fractured in the mesiobuccal and distal root canals, 5 mm to the apical foramen coronally, as per Terauchi et al. [22]. The root canal systems were then randomly distributed (Epidat 4.1, Galicia, Spain) into one of two removal techniques: Group A: Ultrasonic tips (ET25; Satelec Corp, Merignac Cedex, France) (*n* = 20) (US) or Group B: Endo Rescue (Komet Medical, Lemgo, Germany) (*n* = 20) (ER). A bilateral Student’s *t*-test was used with two independent samples to achieve a power of 80.00% for evaluating differences from the null hypothesis H₀: μ₁ = μ₂; factoring in the significance level of 5.00, 40 fractured NiTi endodontic rotary files were needed. The fractured NiTi alloy endodontic rotary files in the US study group were removed using the technique outlined by Ruddle et al., which uses fine ultrasonic tips (ET25; Satelec Corp, Merignac, France) in a counterclockwise motion to remove 1–1.5 mm of dentin around the coronal surface of the fractured file [15]. The obstruction begins to loosen and start spinning during this ultrasonic motion. The ultrasonic generator (Newtron P5, Satelec Corp) power was set to 6 (Figure 1A).

The fractured NiTi endodontic rotary files in the ER study group were removed using the Endo Rescue Kit (Komet Medical, Lemgo, Germany). First, dentin was removed in order to enlarge the root canal entrance using an endodontic bur (H269GK.315.016, Komet Medical, Lemgo, Germany) at a speed of 100,000 rpm. The canal curvature of the coronal root canal third was then straightened, using axial movements with a stainless-steel Gates-Glidden reamer (G180A.204.110, Komet Medical, Lemgo, Germany). Subsequently, a second stainless-steel Gates-Glidden reamer (G180A.204.090, Komet Medical, Lemgo, Germany) was used to create direct access to the fractured file (Protaper Gold, Dentsply Maillefer, Ballaigues, Switzerland). The coronal surface of the files was subsequently exposed by drilling around (RKP. 204.090, Komet Medical, Lemgo, Germany) them at 300 rpm. Finally, the file (Protaper Gold, Dentsply Maillefer, Ballaigues, Switzerland) was removed using the Endo Rescue trepan bur (RKT.204.090, Komet Medical, Lemgo, Germany) at 300 rpm in a counterclockwise movement (Figure 1B).

The NiTi alloy endodontic rotary files (Protaper Gold, Dentsply Maillefer, Ballaigues, Switzerland) in both the US and ER study groups were removed under magnification (OPMI Pico, Zeiss Dental Microscope, Oberkochen, Germany). The time it took to remove the files (Protaper Gold, Dentsply Maillefer, Ballaigues, Switzerland) was recorded in both the US and ER study groups in cases where the files were successfully removed from the root canal system. The working time for removal was established as 90 min for both groups [6].

The teeth were subsequently kept in an incubator (mco-18aic, Sanyo, Moriguchi, Osaka, Japan) (37 °C, 100% relative humidity). The same clinician, who has 10 year’s experience in endodontics, performed all the root canal procedures.

### 2.3. Micro-CT Scanning

A micro-CT scan (Micro-CAT II, Siemens Preclinical Solutions, Knoxville, TN, USA) was performed pre- and postoperatively to analyze the volumetric variations in the amount of dentin removed after root canal procedures to extricate the fractured files (Protaper Gold, Dentsply Maillefer, Ballaigues, Switzerland). The scan was performed using the following exposure parameters: 88 µA, 360° rotation, 90 kV, and isotropic resolution of 50 μm. The 3D tomographic images of the entire tooth had a total of 512 slices, with isotropic 50-micron voxels and a 512 × 512 resolution, according to a previous study [23].

### 2.4. Measurement Procedure

Volumetric analysis of the dentin removed in the distal and mesiobuccal root canal systems subsequent to root canal procedures was performed using image processing software (ImageJ, National Institutes of Health, Bethesda, MD, USA) after identifying and segmenting the fractured NiTi endodontic rotary files (Protaper Gold, Dentsply Maillefer, Ballaigues, Switzerland), as well as the distal and mesiobuccal root canal systems (ROI: 10 × 10 × 10 mm) established using the preoperative and postoperative micro-CT scans (Micro-CAT II, Siemens Preclinical Solutions, Knoxville, TN, USA) (Figure 2).

Next, the teeth were reconstructed, with a 25-micron resolution per voxel (Quantum 3.0, San Jose, CA, USA). An advanced image segmentation technique based on partial differential equations (Level Sets, National Institutes of Health, Bethesda, MD, USA) was then used to divide mesiobuccal and distal root canal systems, enabling progressive differentiation between neighboring pixels and assessment of the anatomy of the root canal systems. The algorithm was initiated manually in the first slice of the volume’s axial view, in which the user traces a contour closely around the channel. With the segmentation technique method, this contour is then deformed towards the inside until it converges, i.e., when the first slice of the root canal system is segmented. Next, the calculated contour was expanded by 6 pixels to initiate the next slice, to which the segmentation technique is once again applied. This process was then applied to each axial slice until the entire channel was 3D segmented. Finally, the difference in the volume of dentin subsequent to removal of the files was calculated in the coronal, medial, and apical third (Figure 3).

### 2.5. Statistical Tests

The studied variables were recorded for statistical analysis (SPSS 22.00, Microsoft Inc., Redmond, WA, USA). Statistical analysis of quantitative variables was performed out using the mean, median, and standard deviation (SD). A comparative analysis was carried out by evaluating the difference between preoperative and postoperative values using the Student′s *t*-test for independent samples or the non-parametric Mann–Whitney–Wilcoxon test, depending on which test they best fit the criteria for. Statistical significance was set as *p* < 0.05.

## 3. Results

Table 1 shows the means and SD values of the preoperative and postoperative differences in dentin volume (mm^3^) between the US and ER study groups at the coronal, medial, and apical level.

The paired *t*-test found no statistically significant differences (*p* = 0.9109) in changes in volume between the US (1.75 ± 1.70 mm^3^) and ER (1.43 ± 0.89 mm^3^) study groups in the mesiobuccal root canal system. In addition, there were no statistically significant differences (*p* = 0.8669) in changes in volume between the US (2.06 ± 1.68 mm^3^) and ER (1.94 ± 1.50 mm^3^) study groups in the distal root canal system (Table 1).

No statistically significant differences (*p* = 0.9109) were found in changes in volume in the coronal root third between the US (0.77 ± 0.96 mm^3^) and ER (0.57 ± 0.58 mm^3^) study groups in the mesiobuccal root canal system. In addition, there were no statistically significant differences (*p* = 0.8814) in changes in volume in the distal root canal system (Figure 4).

No statistically significant differences (*p* = 0.3232) were found in the changes in volume at the medial root third between the US (0.56 ± 0.47 mm^3^) and ER (0.39 ± 0.21 mm^3^) study groups in the mesiobuccal root canal system. In addition, there were no statistically significant differences (*p* = 0.8447) in the change in volume between the US (0.61 ± 0.45 mm^3^) and ER (0.65 ± 0.47 mm^3^) study groups in the distal root canal system (Figure 5).

Finally, no statistically significant differences (*p* = 0.5990) were found in changes in volume at the apical root third between the US (0.43 ± 0.35 mm^3^) and ER (0.50 ± 0.30 mm^3^) study groups in the mesiobuccal root canal system. In addition, there were no statistically significant differences (*p* = 0.6592) in the change in volume between the US (0.53 ± 0.44 mm^3^) and ER (0.45 ± 0.27 mm^3^) study groups in the distal root canal system (Figure 6).

The time it took to remove the NiTi alloy endodontic rotary files randomly assigned to the ER study group ranged between 9–25 min, and 18–90 min for the files assigned to the US group.

Finally, the relationship between the dentin removal and the working time of the removal systems was also analyzed; however, no statistically significant differences were shown at the messiobuccal root canal system (*p* = 0.0252) (Figure 7A) neither the distal root canal system (*p* = 0.0116) (Figure 7B).

## 4. Discussion

The results of the present study rejected the null hypothesis (H_0_) that there is no difference in ease of removal of fractured NiTi alloy endodontic rotary files between ultrasonic tips and the Endo Rescue appliance; however, the present study does accept the null hypothesis (H_0_) that there is no difference in the volume of dentin removal between ultrasonic tips and the Endo Rescue appliance.

NiTi alloy endodontic rotary files removal systems whose removal techniques were different were selected, to analyze the removal capability and dentin removal of each removal system. This is the reason why the ultrasonic tips and the Endo Rescue appliance were used in this study. As these NiTi alloy endodontic rotary files removal systems and techniques have only been recently developed, this highlights that there is still no standardized procedure for safe removal of fractured instruments. A trephine removal technique, using ultrasonics or trepan burs, in combination with a grabbing technique, such as the Masseran Extractor or the Instrument Removal System (Dentsply Tulsa Dental, Johnson City, TN, USA) has been widely used. However, these techniques have shown some limitations and may potentially lead to a weakening of the remaining root [16]. In addition, there are many variables correlated with the success rate of removal of fractured instruments, but these are mainly related to root canal system anatomy in terms of ratio and curvature angle [24,25]. Unexpected instrument fracture often occurs in narrow and curved canals, particularly in mesial root canal systems of lower molars and mesiobuccal root canal systems of upper molars [21]. Furthermore, in the present study, all files were fractured in the mesiobuccal root canal system, similar to a previous study [22], but distal root canal systems were also included in the study to assess the effect of root canal system anatomy on the ease of removal of fractured NiTi alloy endodontic rotary files, compared with the ease of removal of files fractured in the mesial root canal systems. It has also been reported in the literature that NiTi alloy endodontic rotary files tend to fracture more at the midpoint of root canal curvature [26]. However, even though the samples were matched between groups, there could still be some variations, for example, in isthmus size or other interconnections within the main root canal system [1]. The original canal shape should ideally be preserved as much as possible throughout the process of cleaning and shaping that are part of the root canal treatment, as root canal system enlargement of up to 40 to 50% of the root’s width can increase susceptibility to vertical fracture [27,28]. Furthermore, Nevares et al. reported that a digital optical microscope (DOM) made it easier to visualize the fractured files [29]; therefore, an operating microscope was used to enable the visualization of all fragments throughout the removal procedures.

Root perforations were observed in three of the roots randomly assigned to the US group, while no root perforations were identified in the ER group. This could be due to the vibrating tip leading to excessive cutting of dentin tissue. The ultrasonic tip used for the present study was the ET 25, one of the most commonly used ultrasonic tips for removal of fractured NiTi alloy endodontic rotary files [30] due to its small diameter (0.3 mm) and low taper of 3%. The diameter of the root canal system after inserting the ultrasonic tip up to 1 mm was 1.16 mm; the diameter of the root canal system after trephining dentin with the Endo Rescue system was 0.7 mm. A previous study found that the minimal remaining thickness in mandibular molars was 0.60 mm in an ultrasonic study group and 0.66 mm in a trepan bur study group for the fragment located 5 mm below the root canal entrance [1]; however, the present study found no statistically significant differences in the remaining thickness of dentin between both study groups. The ER system was able to remove three fractured NiTi alloy endodontic rotary instruments, while the US technique removed eight fractured NiTi alloy endodontic rotary instruments. However, it is worth mentioning that operator skill and experience may also influence end results.

Suter et al. recommended not taking more than 45 to 60 min to remove fractured NiTi alloy endodontic rotary instruments, because success rates may drop as treatment time increases [12]. They attributed this lower success rate to operator fatigue or over-enlargement of the root canal system, which can compromise tooth integrity and increase the risk of root perforation. Time is generally recorded from the starting straight-line access preparation until the instrument has been successfully removed [1]. For the present study, treatment time was defined as being from beginning to trephine the dentin around the fragment until the instrument was successfully removed or the time limit exceeded. The time needed to remove files in the ER group ranged from 9 to 25 min, while the US group ranged from 18 min to over 1 h. Nagai et al. reported that the time required to remove fractured NiTi alloy endodontic rotary files using an ultrasonic technique ranged from 3 to 40 min, and the time needed to remove fractured NiTi alloy endodontic rotary files using the Masseran technique ranged from 20 min to several hours [31]. In the present study, treatment time was shorter in the trepan bur group than in the ultrasonic group. Therefore, the small-diameter trepan bur technique is recommended for removing instruments that have been fractured coronally or at the curvature of root canals.

Removing separated endodontic rotary files from curved canals poses a challenge for clinicians since these fragments tend to become blocked outside the wall of a curved canal, often causing the retrieval process to be unsuccessful. Therefore, the approach to removing the separated fragments should always include a combination of new techniques and devices that are most likely to be successful, while also minimizing the amount of dentin volume lost and length of treatment time [29]. Therefore, this type of study that assesses and compares the ease of removal of fractured NiTi alloy endodontic rotary files from the root canal system between two different removal systems in straight and curved root canal systems, as well as the volume of dentin removed, can help clinicians select a more effective extraction system for fractured NiTi alloy endodontic rotary files removal.

The standardization of the root canal system anatomy, in ex vivo studies, is a limitation. However, randomization of the sample was carried out. In addition, in future studies, the authors recommend clinical studies with more removal systems.

## 5. Conclusions

Within the limitations of this in vitro study, the results indicate that ultrasonic tips enable greater ease of removal of fractured NiTi endodontic rotary files from the root canal system, with little to no difference in the amount of dentin removed.

## Figures and Tables

**Figure 1 ijerph-19-00718-f001:**
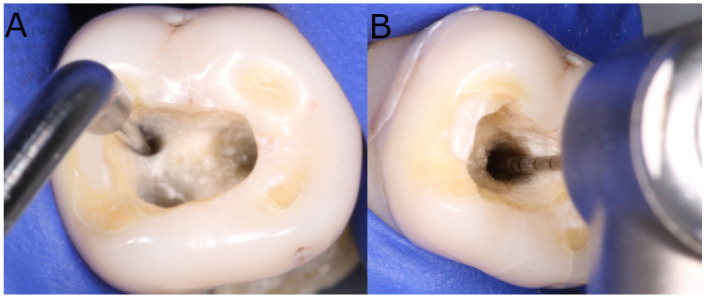
Removal of NiTi endodontic rotary files using (**A**) ultrasonic tip and (**B**) Endo Rescue appliance.

**Figure 2 ijerph-19-00718-f002:**
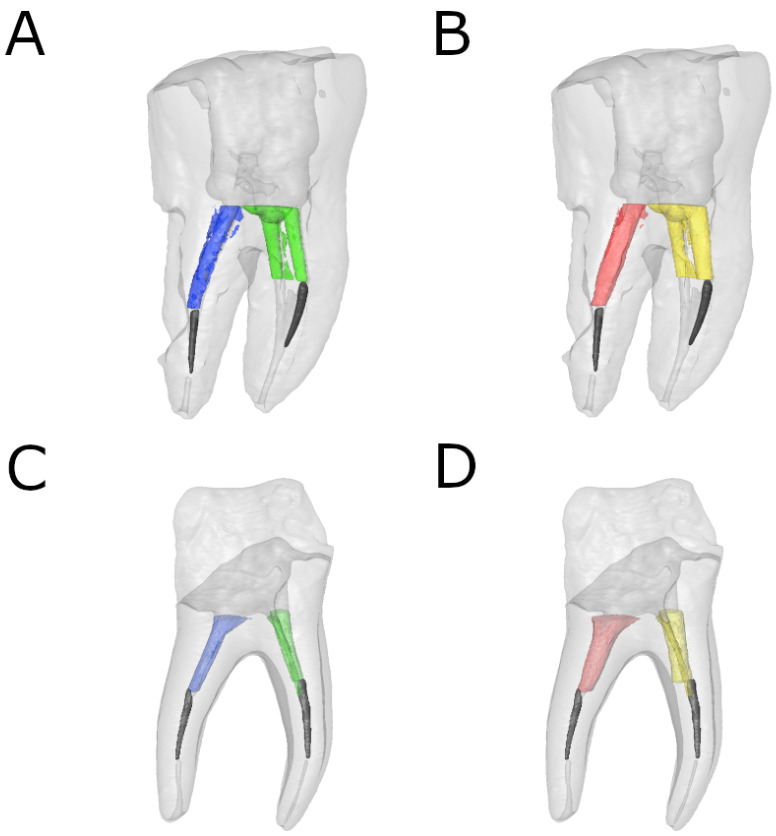
Reconstructed 3D micro-CT images of (**A**) preoperative and (**B**) postoperative ER study group and (**C**) preoperative and (**D**) postoperative US study group. Preoperative (blue) and postoperative (red) distal root canal and preoperative (green) and postoperative (yellow) mesial root canals were isolated. The fractured NiTi endodontic rotary files (black) were also isolated in both distal and mesiobuccal root canal systems.

**Figure 3 ijerph-19-00718-f003:**
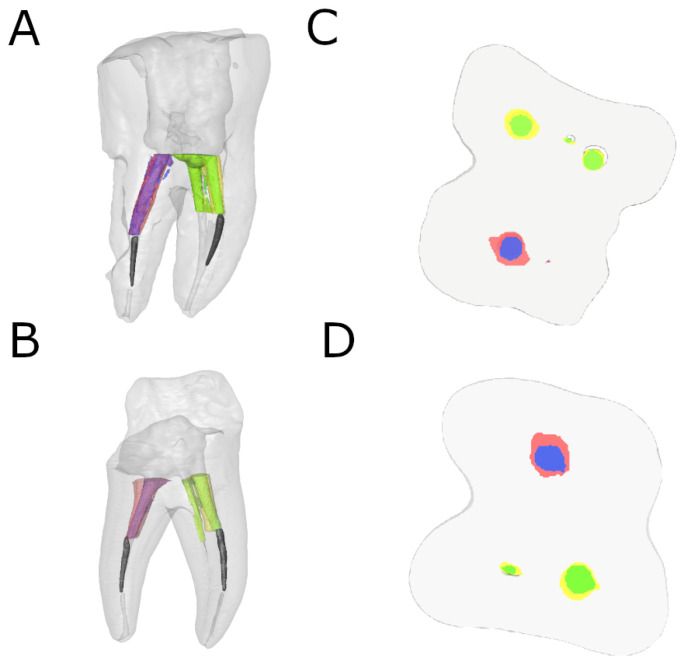
Reconstructed 3D micro-CT images of (**A**) preoperative and postoperative alignment and (**B**) cross-section of ER study group and reconstructed 3D micro-CT images of (**C**) preoperative and postoperative alignment and (**D**) cross-section of US study group. Preoperative (blue) and postoperative (red) distal root canal and preoperative (green) and postoperative (yellow) mesial root canals were aligned. The fractured NiTi endodontic rotary files (black) were also isolated in both distal and mesiobuccal root canal systems.

**Figure 4 ijerph-19-00718-f004:**
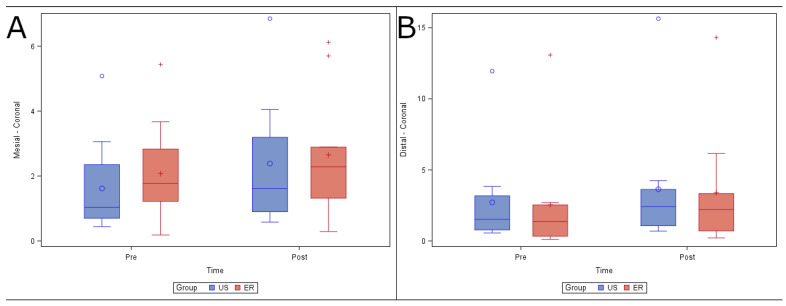
(**A**) Box plot of volume of dentin removed preoperatively and postoperatively, comparing the US and ER study groups at the coronal level in the mesiobuccal and (**B**) distal root canal systems.

**Figure 5 ijerph-19-00718-f005:**
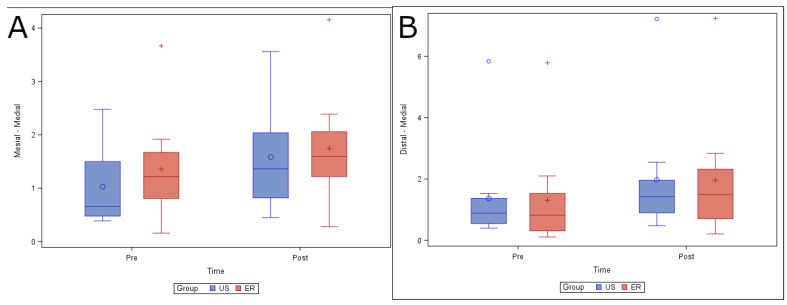
(**A**) Box plot of volume of dentin removed preoperatively and postoperatively, comparing the US and ER study groups at the medial level in the mesiobuccal and (**B**) distal root canal systems.

**Figure 6 ijerph-19-00718-f006:**
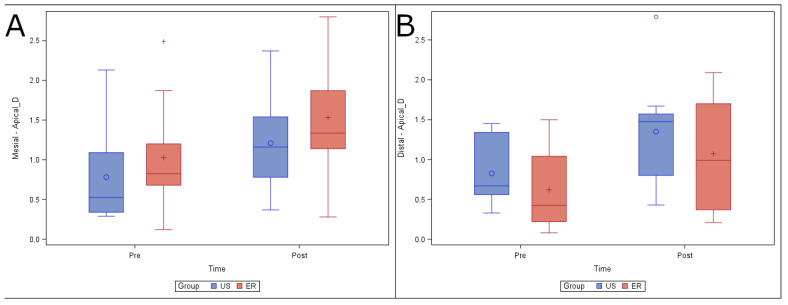
(**A**) Box plot of volume of dentin removed preoperatively and postoperatively, comparing the US and ER study groups at the apical level in the mesiobuccal and (**B**) distal root canal systems.

**Figure 7 ijerph-19-00718-f007:**
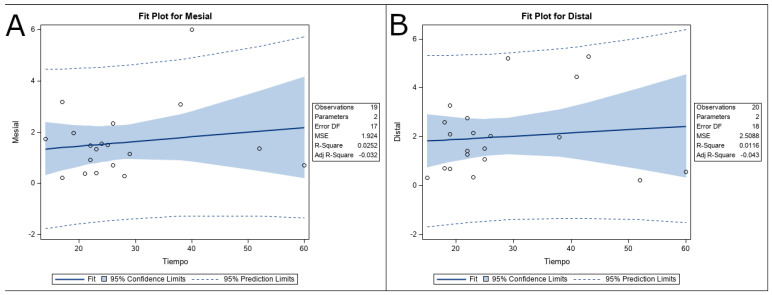
(**A**) Fit plot of the relationship between the dentin removal and the working time of the removal systems at the messiobuccal and (**B**) distal root canal systems. In summary, no differences in overall change in dentin volume were observed between the US and ER study groups after dentin removal. The most important result is that the US study group allowed the removal of 8 out of 20 fractured NiTi endodontic rotary files (1 in the mesiobuccal root canal system and 7 in the distal root canal system) from the root canal system; meanwhile, the ER study group allowed the removal of 3 out of 20 fractured NiTi endodontic rotary files (all in the distal root canal system) from the root canal system. Therefore, the US removal system is recommended for the removal of fractured files from the root canal system.

**Table 1 ijerph-19-00718-t001:** Descriptive values of the preoperative and postoperative volumetric differences (mm^3^) between the US and ER study groups at the coronal, medial, and apical level.

Study Group	Root	Root Third	Time	*n*	Mean	SD	Minimum	Maximum
US	Mesial	Coronal	Pre-op	10	1.62 ^a^	1.48	0.44	5.08
			Post-op	10	2.38 ^a^	1.94	0.58	6.85
		Medial	Pre-op	10	1.03 ^a^	0.76	0.39	2.48
			Post-op	10	1.59 ^a^	0.95	0.45	3.56
		Apical	Pre-op	10	0.78 ^a^	0.59	0.29	2.13
			Post-op	10	1.21 ^a^	0.62	0.37	2.37
	Distal	Coronal	Pre-op	10	2.71 ^a^	3.42	0.56	11.94
			Post-op	10	3.64 ^a^	4.39	0.69	15.64
		Medial	Pre-op	10	1.37 ^a^	1.62	0.40	5.84
			Post-op	10	1.98 ^a^	1.94	0.48	7.22
		Apical	Pre-op	10	0.83 ^a^	0.42	0.33	1.45
			Post-op	10	1.35 ^a^	0.66	0.43	2.79
ER	Mesial	Coronal	Pre-op	10	2.08 ^a^	1.57	0.18	5.44
			Post-op	10	2.65 ^a^	1.90	0.29	6.12
		Medial	Pre-op	10	1.36 ^a^	0.97	0.16	3.67
			Post-op	10	1.75 ^a^	1.03	0.28	4.16
		Apical	Pre-op	10	1.03 ^a^	0.71	0.12	2.49
			Post-op	10	1.53 ^a^	0.77	0.28	2.80
	Distal	Coronal	Pre-op	10	2.54 ^a^	3.83	0.11	14.31
			Post-op	10	3.38 ^a^	2.21	0.21	14.31
		Medial	Pre-op	10	1.31 ^a^	1.69	0.11	5.79
			Post-op	10	1.96 ^a^	2.04	0.21	7.24
		Apical	Pre-op	10	0.62 ^a^	0.52	0.08	1.50
			Post-op	10	1.07 ^a^	0.71	0.21	2.09

US: ultrasonic tips; ER: Endo Rescue ^a^: statistical significance.

## Data Availability

Data available on request due to restrictions eg privacy or ethical.

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
