# Peer review of "Comparative Analysis of Ease of Removal of Fractured NiTi Endodontic Rotary Files from the Root Canal System—An In Vitro Study"

_ijerph, 2022, doi:10.3390/ijerph19020718_

Round 1

Reviewer 1 Report

Nice paper, very interesting. I suggest shortening the introduction ( delete line from 41 to 51). I rather prefer to have more information about figures 2 and 3 (i.e. color significance etc.) I think is very difficult to standardize the selected roots.

Author Response

Dear Reviewer 1:

I’m pleased to resubmit the manuscript of the work entitled, “Comparative Analysis of Ease of Removal of Fractured NiTi Endodontic Rotary Files from the Root Canal System. An In Vitro Study”

Reviewer 1: I don't feel qualified to judge about the English language and style

Response: In order to adapt to the reviewer's 1 comments, we have sent the manuscript to the English Editing Service of MDPI. We attached the Certificate.

Reviewer 1: I suggest shortening the introduction ( delete line from 41 to 51).

Response: In order to adapt to the reviewer's 1 comments, we have deleted lines from 41 to 51.

Reviewer 1: I rather prefer to have more information about figures 2 and 3 (i.e. color significance etc.)

Response: In order to adapt to the reviewer's 1 comments, we have provided more information about Figures 2 and 3.

Reviewer 1: I think is very difficult to standardize the selected roots

Response: In order to adapt to the reviewer's 1 comments, we have added a sentence in the Discussion section as a limitation of the study.

We take this opportunity to thank the recommendations and suggestions made by the reviewers to improve the document.

Yours sincerely,

Reviewer 2 Report

Dear Authors,

Thank you for submitting such intersting and valuable work. The use of rotary files contributes to a great extent to the simplification and long-term predictability of endodontic treatments. However, the fracture of rotary files is every endodontist's worse fear, perhaps alongside perforations. Therefore, any research on realiable methods of broken files removal is of great value. Nevertheless, there are some suggestions that I would made in order to improve the paper's scientific accuracy and maximize its impact:

  • Please justify and explain the choice of the selected two removal file systems (US and ER)
  • Please rephrase the purpose of the study, in accordance with the null hypothesis
  • Was the single operator performing the access cavitites calibrated? In what manner?
  • Were all removal procedures in both groups perforemd by the same operator, as well as access cavity?
  • Please expand explanations to figure 2 and 3 - colour code
  • Please discuss on possible clinical implications of the study
  • Please expand limitations of the study

We look forward to receiving the revised version of your manuscript.

Kind regards!

Author Response

Dear Reviewer 2:

I’m pleased to resubmit the manuscript of the work entitled, “Comparative Analysis of Ease of Removal of Fractured NiTi Endodontic Rotary Files from the Root Canal System. An In Vitro Study”

Reviewer 2: English language and style are fine/minor spell check required

Response: In order to adapt to the reviewer's 2 comments, we have sent the manuscript to the English Editing Service of MDPI. We attached the Certificate.

Reviewer 2: Please justify and explain the choice of the selected two removal file systems (US and ER)

Response: In order to adapt to the reviewer's 2 comments, we have added a sentence in the Discussion section justifying the selection of the removal systems.

Reviewer 2: Please rephrase the purpose of the study, in accordance with the null hypothesis

Response: In order to adapt to the reviewer's 2 comments, we have rephrase the purpose of the study in the Abstract and the Introduction sections.

Reviewer 2: Was the single operator performing the access cavitites calibrated? In what manner?

Response: In order to adapt to the reviewer's 2 comments, we clarify that the same clinician, who has 10 years’ experience in endodontics, performed all the root canal procedures, including the access cavities. The single operator was trained and calibrated using the kappa index (Cohen J. A coefficient of agreement for nominal scales. Educ Psychol Meas 1960; 20: 37-46.), obtaining values between 0.61-0.69, which is considered substantial according to the Landis and Koch scale (Landis JR, Koch GG. The measurement of observer agreement for categorical data. Biometrics 1977; 33(1):159-74.).

Reviewer 2: Were all removal procedures in both groups perforemd by the same operator, as well as access cavity?

Response: In order to adapt to the reviewer's 2 comments, we clarify that the same clinician, who has 10 years’ experience in endodontics, performed all the root canal procedures, including the removal procedures.

Reviewer 2: Please expand explanations to figure 2 and 3 - colour code

Response: In order to adapt to the reviewer's 2 comments, we have provided more information about Figures 2 and 3.

Reviewer 2: Please discuss on possible clinical implications of the study

Response: In order to adapt to the reviewer's 2 comments, we clarify that this type of study that assess and compare the ease of removal of fractured NiTi alloy endodontic rotary files from the root canal system between two different removal systems in straight and curved root canal systems, as well as the volume of dentin removed, can help the clinician to select the more effective extraction system for fractured NiTi alloy endodontic rotary files removal. We have added this sentence in the Discussion section.

Reviewer 2: Please expand limitations of the study

Response: In order to adapt to the reviewer's 2 comments, we have added the limitations of the study in the Discussion section.

We take this opportunity to thank the recommendations and suggestions made by the reviewers to improve the document.

Yours sincerely,

Reviewer 3 Report

The study is well designed and well executed. Concepts are explained correctly according to the current understanding in the field.Terminology is defined and used in a consistent and acceptable way.

After reading the manuscript I have some doubts and I will make some considerations

1: I consider that the authors should emphasize that with the Ultrasonic tips 8 files out of 20 were rescued and with the Endo Rescue 3 out of 20, I think that the difference is substantial and this should be the most important result.

2: Is it the same professional who removes the rotatory file with both systems? If so, does he have the same experience in handling both systems? We all know the importance of the learning curve. Operator skill and experience can influence completely the results.

3: in table 1, which means n10, you use 20 samples with each System don’t you?

4: I understand that the analysis of dentin loss has been performed on the entire sample or only on those in which the file was removed. It would be good to make it clear

5: Have you study the relation between the time spent on removing the fractured files from the root canal and the amount of dentin removed?. it is quite possible that there is a relationship between time spent and dentin loss, which could confound the results.

Author Response

Dear Reviewer 3:

I’m pleased to resubmit the manuscript of the work entitled, “Comparative Analysis of Ease of Removal of Fractured NiTi Endodontic Rotary Files from the Root Canal System. An In Vitro Study”

Reviewer 3: I don't feel qualified to judge about the English language and style

Response: In order to adapt to the reviewer's 3 comments, we have sent the manuscript to the English Editing Service of MDPI. We attached the Certificate.

Reviewer 3: I consider that the authors should emphasize that with the Ultrasonic tips 8 files out of 20 were rescued and with the Endo Rescue 3 out of 20, I think that the difference is substantial and this should be the most important result.

Response: In order to adapt to the reviewer's 3 comments, we have better explained that the US study group allowed removing  8 out of 20 fractured NiTi endodontic rotary files (1 in the messiobuccal root canal system and 7 in the distal root canal system) removed from the root canal system; meanwhile, the ER study group allowed removing 3 out of 20 fractured NiTi endodontic rotary files (all in the distal root canal system) removed from the root canal system and we have also have emphasized the most important result of the study in the Results section.

Reviewer 3: Is it the same professional who removes the rotatory file with both systems? If so, does he have the same experience in handling both systems? We all know the importance of the learning curve. Operator skill and experience can influence completely the results.

Response: In order to adapt to the reviewer's 3 comments, we clarify that the same clinician, who has 10 years’ experience in endodontics, performed all the root canal procedures and had the same experience in handling both removal systems.

Reviewer 3: in table 1, which means n10, you use 20 samples with each System don’t you?

Response: In order to adapt to the reviewer's 3 comments, we clarify that we removed 20 NiTi alloy endodontic rotary files with each system: 10 NiTi alloy endodontic rotary files in the mesial root canal system and 10 NiTi alloy endodontic rotary files in the distal root canal system

Reviewer 3: I understand that the analysis of dentin loss has been performed on the entire sample or only on those in which the file was removed. It would be good to make it clear

Response: In order to adapt to the reviewer's 3 comments, we clarify that the analysis of dentine removal was performed in those in which the files were removed: distal and messiobuccal root canal systems. The authors have added a sentence in the Material and Methods section.

Reviewer 3: Have you study the relation between the time spent on removing the fractured files from the root canal and the amount of dentin removed?. it is quite possible that there is a relationship between time spent and dentin loss, which could confound the results.

Response: In order to adapt to the reviewer's 3 comments, we clarify that the relationship between the dentin removal and the working time of the removal systems was also analyzed; however, no statistically significant differences were shown at the messiobuccal root canal system (p = 0.0252) neither the distal root canal system (p = 0.0116). The authors have added a sentence and illustration (Figure 7A and B) in the Results section.

We take this opportunity to thank the recommendations and suggestions made by the reviewers to improve the document.

Yours sincerely,

Reviewer 4 Report

-A more detailed description of the Endo Rescue kit is required

-In the results part it is not clear to me if all the fragments have been removed from the canals. If not, the reasons for the failure should be better specified.

-Considering the great difference between the canal morphology of the distal and mesial canal of the lower molar, clinically there is a great difference in the removal of an instrument from the distal canal or the mesial canal. This has a great impact both on the possibility of removal regardless of the technique and in the time taken to remove.

Author Response

Dear Reviewer 4:

I’m pleased to resubmit the manuscript of the work entitled, “Comparative Analysis of Ease of Removal of Fractured NiTi Endodontic Rotary Files from the Root Canal System. An In Vitro Study”

Reviewer 4: I don't feel qualified to judge about the English language and style

Response: In order to adapt to the reviewer's 4 comments, we have sent the manuscript to the English Editing Service of MDPI. We attached the Certificate.

Reviewer 4: A more detailed description of the Endo Rescue kit is required

Response: In order to adapt to the reviewer's 4 comments, we have provided a more detailed description of the Endo Rescue kit in the Material and Methods section.

Reviewer 4: In the results part it is not clear to me if all the fragments have been removed from the canals. If not, the reasons for the failure should be better specified.

Response: In order to adapt to the reviewer's 4 comments, we clarify that the US study group allowed removing  8 out of 20 fractured NiTi endodontic rotary files (1 in the messiobuccal root canal system and 7 in the distal root canal system) removed from the root canal system; meanwhile, the ER study group allowed removing 3 out of 20 fractured NiTi endodontic rotary files (all in the distal root canal system) removed from the root canal system and we have also emphasized the most important result of the study in the Results section.

Reviewer 4: Considering the great difference between the canal morphology of the distal and mesial canal of the lower molar, clinically there is a great difference in the removal of an instrument from the distal canal or the mesial canal. This has a great impact both on the possibility of removal regardless of the technique and in the time taken to remove.

Response: In order to adapt to the reviewer's 4 comments, we clarify that the reason of including the mesial and distal root canal systems was to provide useful information to the clinician for selecting the most effective removal system according to the location of the fractured NiTi alloy endodontic rotary files.

We take this opportunity to thank the recommendations and suggestions made by the reviewers to improve the document.

Yours sincerely,